# Net-Patterned Fluorine-Doped Tin Oxide to Accelerate the Electrochromic and Photocatalytic Interface Reactions

**Seock-Joon Jeong †, Kue-Ho Kim † and Hyo-Jin Ahn \***

A Department of Materials Science and Engineering, Seoul National University of Science and Technology, Seoul 01811, Korea; 16100585@seoultech.ac.kr (S.-J.J.); kueho0131@seoultech.ac.kr (K.-H.K.)

\* Correspondence: hjahn@seoultech.ac.kr; Tel.: +82-02-970-6622

† S.-J. Jeong and K.-H. Kim contributed equally to this work.

**Abstract:** In this study, the surface morphology of net-patterned fluorine-doped tin oxide (FTO) films was optimized with mesh sizes (60 mesh, 40 mesh, and 24 mesh) using the one-pot horizontal ultrasonic spray pyrolysis deposition (HUSPD) process. The 40M-FTO sample exhibited optimized electrical and optical properties due to the improved crystallinity and net-patterned surface morphology of FTO. The electrochromic (EC) electrodes fabricated with 40M-FTO showed superior EC performance, including transmittance modulation ($\Delta T$, 58.7%), switching speeds (4.1 s for coloration and 5.9 s for bleaching), and coloration efficiency (CE, 52.4 $cm^2$/C). These optimum values were attributed to the combined effect of the enhanced electrical properties from the improved crystallinity of the $SnO_2$ and the high transmittance with a large surface area stemming from the optimization of the net-patterned FTO surface morphology. Moreover, the improved reaction sites with large surface area and enhanced electrical conductivity can facilitate the photocatalytic reaction. Accordingly, we suggest our novel strategy for use in creating promising transparent conducting electrodes that can be fabricated with net-patterned FTO to realize enhanced electrochromic and photocatalytic interface reactions.

**Keywords:** transparent conducting oxide; electrical and optical properties; transition metal oxides; patterning; electrochromic performances





## 1. Introduction

With the advent of electrochromism, electrochromic devices (ECDs) have attracted much attention around the world due to their potential and wide use in areas such as cutting-edge displays, smart windows, and energy conservation devices [1–3]. ECDs can reversibly change their optical properties (transmittance, absorption, and reflectance) from electronic structure variations of electrochromic (EC) films. When an electric field is applied, EC films undergo electrochemical oxidation and a reduction reaction, accompanied by the transport of ions and electrons [4–7]. Typical ECDs are composed of the three main components of EC films (cathodic and anodic), an electrolyte, and transparent conducting oxide (TCO) layers, which can directly affect key performance outcomes of ECDs, such as the coloration efficiency (CE), switching speed, and transmittance modulation [8,9]. Among them, the TCO serves as an electron pathway, which connects the EC films with an external circuit and determines the overall transmittance of the ECD. Moreover, the TCO can be used for photocatalysis application due to its advantageous electronic properties and conductivity [10–13]. For example, Khalaf et al. demonstrated the photocatalytic behavior of Co-doped $TiO_2$, and Jiamprasertboon et al. reported the optimized photocatalytic properties of Cl-doped ZnO films with enhanced electrical and optical properties [14,15]. At this point, enhancing the electrical conductivity and transmittance of the TCO is suggested as one of the most effective strategies for improving the electrochromic and photocatalytic performances. Generally, transition metal oxide films, including tin-doped indium oxide (ITO), fluorine-doped tin oxide (FTO), and aluminum-doped zinc oxide (AZO), have been

used as the TCO layer in ECDs [16–18]. Specifically, ITO has been widely used for broad optoelectronic applications (e.g., touch panels, electroluminescent devices, organic light-emitting diodes, and light sensors) as a typical TCO material due to its superior visible light transmittance (>80%), low resistivity (<$10^{-3}$ Ω cm), and large band gap (~4.8 eV) [19,20]. However, ITO has several drawbacks, including its high element cost, low mechanical strength, and low chemical stability, which considerably limit further applications. FTO was recently proposed as a substitute material due to its low cost and excellent mechanical and chemical durability. In order to achieve high transmittance while maintaining the electrical conductivity of FTO films, various methods have been suggested, including patterning, metal ion doping, and the use of layered structures [21,22]. For example, Lee et al. fabricated micro-patterned FTO films using a photolithography process and applied this method to ECDs with Prussian blue [23]. They reported enhanced electrochromic performance with patterned FTO, such as rapid switching speeds (~30 s) and large optical density levels (>0.4). However, the conventional patterning process is associated with complex procedures, which require a patterned mask, UV light exposure, and the coating and stripping of a photoresist. Therefore, a progressive synthesis method for patterned FTO films with a simple process is needed to lower the manufacturing cost and enhance the electrical and optical properties.

In this study, we report a one-pot synthesis technique for net-patterned FTO films using the horizontal ultrasonic spray pyrolysis deposition (HUSPD) method with various types of stainless meshes. This simple synthesis method can effectively simplify the manufacturing procedure and decrease the time, allowing the realization of affordable net-patterned FTO films. Through an adjustment of the patterning size, the surface area and transmittance of the FTO films, factors that directly affect the EC performance were successfully optimized. To demonstrate this performance enhancement, the correlations of the morphological, structural, optical, and electrical properties and the EC performance outcomes were investigated in detail.

## 2. Results and Discussion

### 2.1. Preparation and Morphological Properties

Figure 1 shows an overall experimental illustration of the net-patterned FTO films created using the one-pot HUSPD process with a stainless mesh. Figure 1a displays an outline of the HUSPD system, which is composed of (1) a reaction furnace, (2) a gas flow controller, (3) a generator, and (4) a spraying bath with an ultrasonic atomizer [24]. The conventional USPD system uses a spraying head directly above the substrate with a vertically supplied precursor solution, which can cause nonuniform film structures. Distinctively, this HUSPD system uses horizontally supplied precursor droplets supplied by an ultrasonic atomizer and a gas flow for the pyrolysis process, which leads to the formation of uniformly coated film structures [25]. The atomized precursor droplets undergo the subsequent process of solvent evaporation, solute condensation, and thermal decomposition (pyrolysis) [26]. Moreover, HUSPD offers the advantage of allowing simple adjustments to experimental parameters, such as the frequency of the atomizer, the substrate rotating speed, and the gas flow rate, among others. Figure 1b shows the manufacturing procedure used to synthesize the net-patterned FTO films. When the stainless mesh covers the glass substrate during the HUSPD process, the pyrolysis reaction occurs more actively on the uncovered area, which can readily generate the net pattern on the substrate. This one-pot synthesis of net-patterned FTO films can effectively decrease the time and cost of the patterning process compared to the photolithography method, which requires a patterned mask, UV exposure, and a photoresist. The net-patterned FTO films possess a large surface area with enhanced surface roughness and high transmittance because less of the area is coated. This ensures a significant EC performance improvement. To adjust the net-pattern size of the FTO glass, various types of meshes (60 mesh, 40 mesh, 24 mesh) were covered on the prepared glass substrate during the HUSPD process. For comparison, the HUSPD process was also

conducted on a glass substrate without a cover mesh (samples are henceforth referred to as bare FTO, 60M-FTO, 40M-FTO, and 24M-FTO).

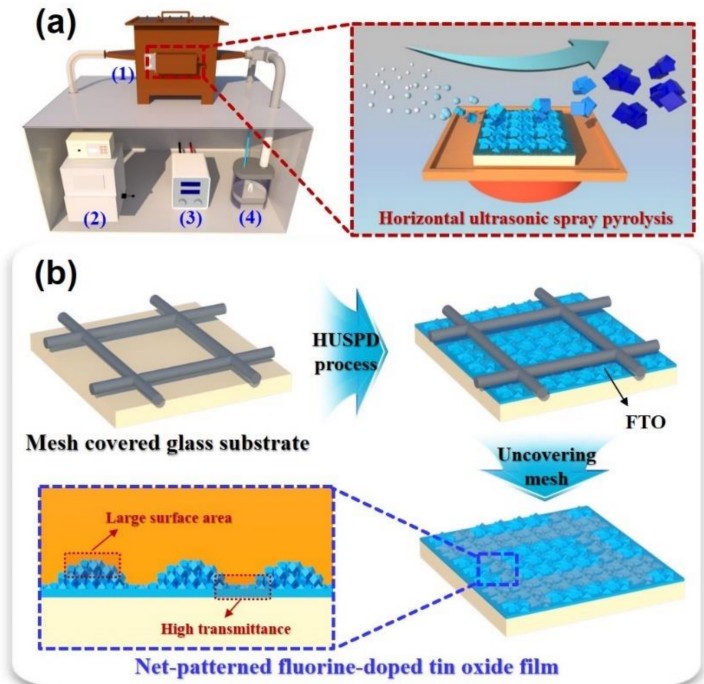

**Figure 1.** Schematic illustration of the procedure for the synthesis of net-patterned fluorine-doped tin oxide (FTO) film, (**a**) outline of the horizontal ultrasonic spray pyrolysis deposition (HUSPD) system, (**b**) manufacturing procedure including HUSPD process on the mesh covered substrate and mesh removal to synthesize the net-patterned FTO films.

Figure 2a,d correspondingly shows top-view optical microscope (OM) images of the bare FTO, 60M-FTO, 40M-FTO, and 24M-FTO samples prepared with different types of meshes. Figure 2a exhibits a smooth and uniform film structure over the entire area. In contrast, Figure 2c,d shows a net-patterned morphology with various patterning sizes (~250 um for 60M-FTO, ~500 um for 40M-FTO, and ~900 um for 24M-FTO). This result is attributed to the different spacing sizes of the respective meshes, which takes a role as a pathway of sprayed precursor droplets during the HUSPD process.

Figure 3 shows the top-view and cross-view field-emission scanning electron microscope (FESEM) images of (a, e) bare FTO, (b, f) 60M-FTO, (c, g) 40M-FTO, and (d, h) 24M-FTO prepared by the HUSPD process, respectively. The average thickness of the FTO films was measured as ~263.4 nm for bare FTO, ~299.0 for 60M-FTO, ~346.5 for 40M-FTO, and ~326.7 nm for 24M-FTO. Moreover, the average crystal size was measured as ~144.2 nm for bare FTO, ~184.1 nm for 60M-FTO, ~204.5 nm for 40M-FTO, and ~226.3 nm for 24M-FTO. The increased film thickness and crystal size of 40M-FTO can be attributed to the carrier concentration and crystallinity enhancement [7,27].

### 2.2. Structral and Chemical Properties

To investigate the surface properties further, a confocal laser scanning microscope (CLSM) analysis was conducted, as this method can profile a 3D surface structure and evaluate the surface area and roughness of films (see Figure 4a). The average depth and height values were increased from 0.115 um for the bare FTO sample to 0.205 um for the 60M-FTO sample and to 0.382 um for the 40M-FTO sample. This result is caused by the net-patterning and increment of the spacing size according to the mesh type. While 60M-FTO showed the largest number of patterns, the spacing area of the 60 mesh is not large enough to accommodate a sufficient amount of precursor droplets. Moreover, 24M-FTO

shows lower average depth and height values than 40M-FTO in spite of the large spacing area, stemming from the deficient number of patterns in this case. The surface area values of all samples also exhibit the same tendency, with average depth and height values of $187.2 \times 10^3$ um$^2$ for the bare FTO sample, $198.5 \times 10^3$ um$^2$ for 60M-FTO, $221.7 \times 10^3$ um$^2$ for 40M-FTO, and $212.6 \times 10^3$ um$^2$ for 24M-FTO. The surface area and roughness were optimized in the 40M-FTO case, which can provide an additional interface between EC films and the TCO layer and facilitate the electrochemical reaction of ECDs [28,29].

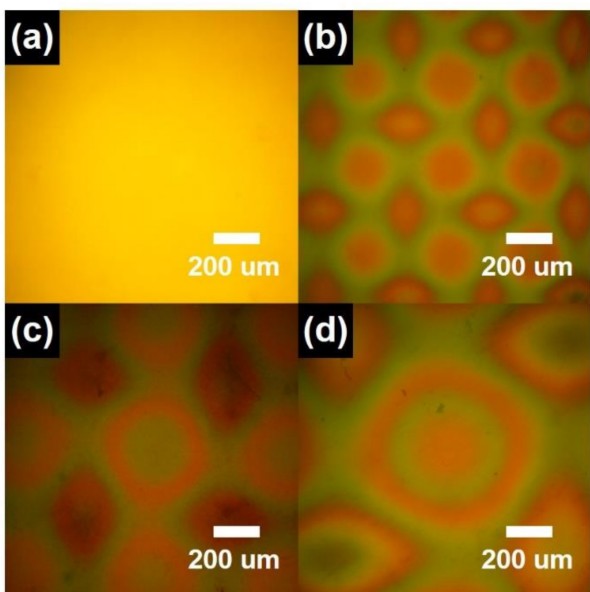

**Figure 2.** (**a–d**) Optical microscope (OM) images of the bare FTO, 60M-FTO, 40M-FTO, and 24M-FTO samples, respectively.

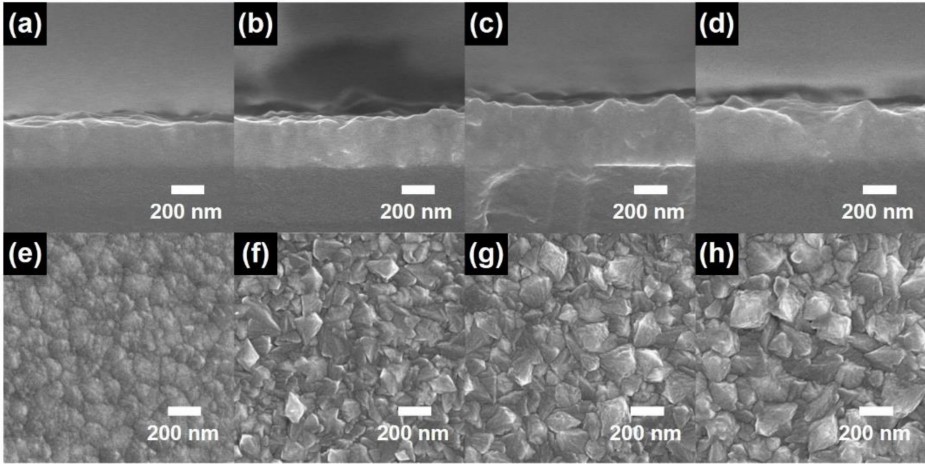

**Figure 3.** Top-view and cross-view FESEM images of (**a**,**e**) bare FTO, (**b**,**f**) 60M-FTO, (**c**,**g**) 40M-FTO, and (**d**,**h**) 24M-FTO, respectively.

To investigate the crystal structure, XRD measurements were taken of the bare FTO, 60M-FTO, 40M-FTO, and 24M-FTO samples (see Figure 4b). All FTO films fabricated by the HUSPD process showed characteristic diffraction peaks at 26.61°, 33.89°, 38.10°, and 51.78°, corresponding to the (110), (101), (200), and (211) planes of the SnO$_2$ phase (JCPDS (Joint Committee on Powder Diffraction Standards) card No. #88-0287) [30]. In the bare FTO case, the lowest crystallinity was detected due to the low pyrolysis temperature of 350 °C and relatively short deposition time compared to the net-patterned FTO films [31]. This enhanced crystallinity of net-patterned FTO films can contribute to important electrical

properties, including the carrier concentration and mobility. Specifically, 40M-FTO showed the highest (200)/(110) plane intensity ratio of 1.25 compared to the value of 0.52 for 60M-FTO and 0.80 for 24M-FTO, which is known to be a preferred structure for electron transfer [32,33]. The improvement of the electrical properties of the TCO layer will strongly affect the switching speeds of ECDs. Therefore, the formation of net-patterned FTO films with the (200) preferred orientations can be successfully achieved using the HUSPD process, which can provide efficient electron transport within the TCO layer.

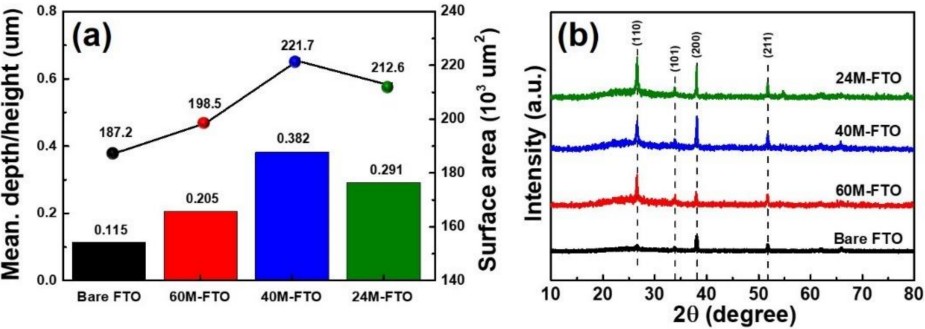

**Figure 4.** (**a**) Measured average depth and height values and surface area of bare FTO, 60M-FTO, 40M-FTO, and 24M-FTO samples and (**b**) XRD curves of bare FTO, 60M-FTO, 40M-FTO, and 24M-FTO samples, implying crystallinity enhancement from bare FTO to 24M-FTO.

Figure 5 shows the electrical and optical properties with the calculated figure of merit for all of the samples. The electrical properties (resistivity, carrier concentration, and Hall mobility) of the samples were examined using a Hall-effect measurement system (see Figure 5a). Interestingly, the carrier concentration and Hall mobility of the net-patterned FTO films were enhanced compared to those of the bare FTO films. As summarized in Table 1, the carrier concentration was improved from $7.82 \times 10^{20}$ cm$^{-3}$ for the bare FTO to $1.02 \times 10^{21}$ cm$^{-3}$ for 60M-FTO and to $1.21 \times 10^{21}$ cm$^{-3}$ for 40M-FTO. The Hall mobility was also increased from 12.8 cm$^2$/(V·s) for the bare FTO to 19.1 cm$^2$/(V·s) for 60M-FTO and to 22.8 cm$^2$/(V·s) for 40M-FTO. These results are ascribed to the enhanced crystallinity and FTO film thickness [34]. However, for 24M-FTO, decreased carrier concentration ($1.14 \times 10^{21}$ cm$^{-3}$) and Hall mobility (21.8 cm$^2$/(V·s)) outcomes were observed due to the reduced value of the (200)/(110) plane ratio, which offers advantages in terms of the electrical properties. Therefore, the 40M-FTO sample exhibited optimum electrical properties, including the lowest resistivity of $4.27 \times 10^{-4}$ Ω·cm. Figure 5b displays the optical transmittance curve of the bare FTO, 60M-FTO, 40M-FTO, and 24M-FTO samples measured in the wavelength range of 300 nm to 800 nm. The bare FTO showed the lowest average transmittance of 80.3% in the visible range (400~800 nm), caused by considerable light scattering due to the dense FTO films with low crystallinity [35]. On the other hand, the net-patterned FTO films showed superior average transmittance outcomes of 89.8%, 88.9%, and 86.2% for 60M-FTO, 40M-FTO, and 24M-FTO, as high transmittance is one of the representative achievements of this patterning method (light penetration through the patterns). The figure of merit ($\varphi = T^{10}/R_{sh}$) was calculated from obtained transmittance ($T$) and sheet resistance values ($R_{sh}$), and 40M-FTO showed the highest value of $3.84 \times 10^{-2}$ Ω$^{-1}$, as expected [36]. Therefore, the electrical and optical properties were optimized in the 40M-FTO case. This combination can therefore represent key factors for better EC performances.

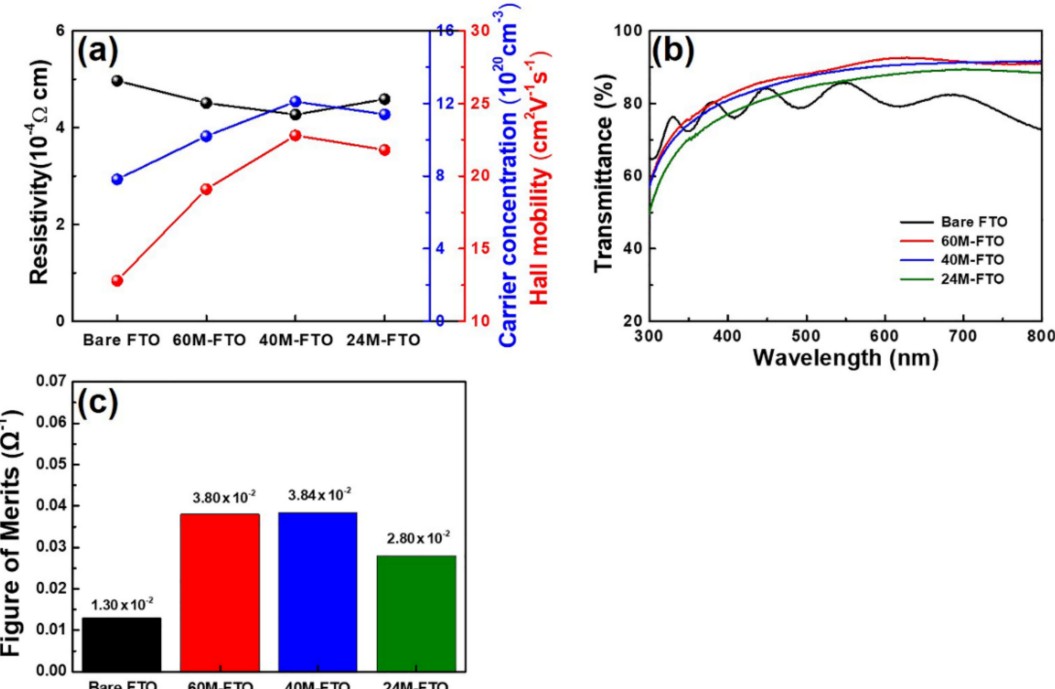

**Figure 5.** Electrical and optical properties of bare FTO, 60M-FTO, 40M-FTO, and 24M-FTO, respectively. (**a**) Electrical properties with carrier concentration, Hall mobility, and resistivity, (**b**) optical transmittance in the wavelength range of 300~800 nm, and (**c**) FOM (Figure of merits) values.

**Table 1.** Summarized electrical and optical data of bare FTO, 60M-FTO, 40M-FTO and 24M-FTO samples.

| Samples | Bare FTO | 60M-FTO | 40M-FTO | 24M-FTO |
|---|---|---|---|---|
| Carrier concentration (cm$^{-3}$) | $7.82 \times 10^{20}$ | $1.02 \times 10^{21}$ | $1.21 \times 10^{21}$ | $1.14 \times 10^{21}$ |
| Mobility (cm$^2$/V·S) | $1.28 \times 10$ | $1.91 \times 10$ | $2.28 \times 10$ | $2.18 \times 10$ |
| Resistivity (Ω·cm) | $4.97 \times 10^{-4}$ | $4.51 \times 10^{-4}$ | $4.27 \times 10^{-4}$ | $4.59 \times 10^{-4}$ |
| Sheet resistance (Ω/□) | $8.39 \pm 0.17$ | $8.98 \pm 0.13$ | $8.03 \pm 0.21$ | $8.11 \pm 0.24$ |
| Transmittance (%) | 80.3 | 89.8 | 88.9 | 86.2 |
| Figure of Merit (Ω$^{-1}$) | $1.30 \times 10^{-2}$ | $3.80 \times 10^{-2}$ | $3.84 \times 10^{-2}$ | $2.80 \times 10^{-2}$ |

*2.3. Electrochemical and Electrochromic Performance*

To demonstrate the enhancing effect of TCO layers on the ECDs, all fabricated FTO films were coated with WO$_3$ film and annealed at 300 °C to form a FTO-WO$_3$ EC electrode. Figure 6 shows the CV (Cyclic voltammetry) curves of all of the EC electrodes fabricated with bare FTO, 60M-FTO, 40M-FTO, and 24M-FTO. During the cycling process, all of the electrodes exhibited a single pair of reduction-oxidation (redox) curves with respective anodic and cathodic peaks. The redox reaction indicates the intercalation and deintercalation of Li ion and electron transfers to the WO$_3$ films, accompanied by a reversible color change from a bleached state (transparent) to a colored state (deep blue). The current density levels of the both the anodic and cathodic peaks were gradually increased from the bare FTO to the 40M-FTO sample. This result implies the enhancement of electrochemical activity due to the additional interface area between the EC film and the TCO layer, causing an increased number of ions and electrons participating in the redox reaction [37]. However, the EC electrode with 24M-FTO showed a slight reduction in the peak current density due to the decreased number of patterns, as expected from the aforementioned surface area and mean depth/height data in the CLSM results.

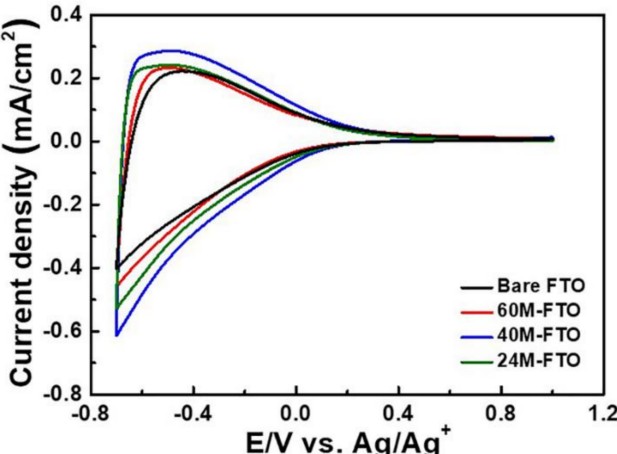

**Figure 6.** CV (cyclic voltammetry) curves of all samples measured between −0.7 and 1.0 V at the scan rate of 20 mV/s using three-electrode system.

Figure 7a presents the in-situ optical transmittance of the fabricated FTO-WO$_3$ EC electrodes at a wavelength of 633 nm and with the potential repeatedly applied at −0.7 V (colored state) and 1.0 V (colored state) to all of the electrodes. As summarized in Table 2, the transmittance modulation (ΔT, transmittance gap between the colored and bleached states), switching speed (response time to attain 90% of the ΔT), and coloration efficiency (CE) of all electrodes were measured. The 40M-FTO sample exhibited the highest ΔT of 58.7%, which is 9.9% higher than that in the bare FTO case. This result is attributed to the increment of the transmittance in the bleached state caused by the optimized net-patterned morphology of the FTO films. The reduced transmittance in the colored state was caused by the facilitated electrochemical activity through the large interfacial area between the EC film and TCO layer. Moreover, shortened switching speeds were noted in the 40M-FTO case (4.1 s for the coloration speed and 5.9 s for the bleaching speed) compared to the bare FTO case (11.0 s for the coloration speed and 11.0 s for the bleaching speed) and the other net-patterned FTO films. The fast switching speed of 40M-FTO is associated with the optimized electrical properties, including the carrier concentration and Hall mobility, boosting the electron and Li ion transport within the ECDs during the electrochemical reaction. Figure 7b shows the calculated CE values of all electrodes, signifying comprehensive EC performance [38,39].

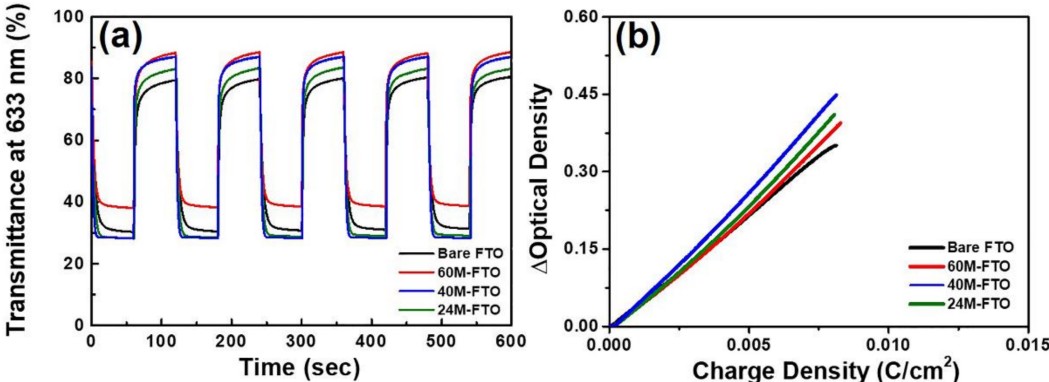

**Figure 7.** (**a**) In situ optical transmittance curves investigated with the repetitive potential at −0.7 V for the colored state and 1.0 V for the bleached state and (**b**) optical density variation by the charge density insertion.

**Table 2.** Summarized electrochromic performance of all the FTO-WO$_3$ electrodes.

| Samples | T$_{bleached}$ (%) | T$_{colored}$ (%) | Modulation (%) | Switching Speed (Colored) | Switching Speed (Bleached) | Coloration Efficiency (cm$^2$/C) |
|---|---|---|---|---|---|---|
| Bare FTO | 80.1 | 31.3 | 48.8 | 11.0 | 11.0 | 38.2 |
| 60M-FTO | 88.3 | 38.3 | 50.0 | 7.5 | 10.3 | 42.6 |
| 40M-FTO | 87.0 | 28.3 | 58.7 | 4.1 | 5.9 | 52.4 |
| 24M-FTO | 83.3 | 31.9 | 54.5 | 6.2 | 9.5 | 47.3 |

The CE value is defined as the optical density (OD) according to the charge densities (*Q/A*). It is calculated using Equations (1) and (2). The CE values were 38.2 cm$^2$/C for the bare FTO, 42.6 cm$^2$/C for the 60M-FTO sample, 52.4 cm$^2$/C for the 40M-FTO sample, and 47.3 cm$^2$/C for the 24M-FTO sample, indicating that 40M-FTO exhibited optimized EC performance. This outcome is attributed to the large value of ΔT from the net-patterning process and the enhanced electrochemical activity from the additional surface area of the FTO films.

$$CE = \Delta OD/(Q/A) \tag{1}$$

$$\Delta OD = \log(T_b/T_c) \tag{2}$$

Thus, superior EC performance was achieved at the electrodes with the 40M-FTO TCO layer, mainly due to the optimized net-patterning process using the one-pot HUSPD method. Firstly, the net-patterned FTO morphology effectively enhanced the surface area and transmittance of the TCO layer, which can affect the ΔT and CE values. Secondly, optimized electrical properties, in this case the carrier concentration and Hall mobility, were realized due to the improved FTO crystallinity. Therefore, using the HUSPD process with various types of stainless meshes can be suggested as a simple and effective method for the patterning of TCO materials onto substrates.

## 3. Materials and Methods

### 3.1. Experimental Details

The net-patterned FTO films were successfully fabricated on glass substrates (Corning, Eagle XGTM) using the one-pot HUSPD (Ceon, TV 500, Hwaseong, Korea) process with a stainless mesh (Goryeo Ltd., SUS 304, Seoul, Korea). First, the glass substrates were cleaned with acetone (CH$_3$COCH$_3$, Aldrich) and ethanol (CH$_3$CH$_2$OH, DUKSAN) using ultrasonication for 25 min and then washed with de-ionized (DI) water. Subsequently, to prepare the precursor solution for the HUSPD process, 0.68 M tin chloride pentahydrate (SnCl$_4$·5H$_2$O, SAMCHUN) and 1.2 M ammonium fluoride (NH$_4$F, JUNSEI) were dissolved into DI water with 5 vol% ethanol. After stirring for 3 h, the solution turned transparent, and this precursor solution was poured into an ultrasonic atomizer (1.6 MHz). The temperature of the pyrolysis chamber, the carrier gas (air) flow rate, and the rotation speed were correspondingly fixed at 350 °C, 15 L/min, and 5 rpm for the HUSPD process. Due to the existence of stainless meshes and the difference of spacing sizes of them, the loading amount of sprayed precursor droplets onto the substrate was varied for each sample. Thus, for comparison, we adjusted the deposition time (15, 35, 30, and 25 min for the bare FTO, 60M-FTO, 40M-FTO, and 24M-FTO) to collect FTO samples with similar sheet resistance (~8 Ω/□).

### 3.2. Characterization

#### 3.2.1. Structures and Morphologies

The surface morphology and roughness were investigated using optical microscopy (OM, Microscopes INC., S39B), field-emission scanning electron microscopy (FESEM, Hitachi S–4800) and a 3D surface confocal laser scanning microscope (CLSM, Carl Zeiss, LSM 800 MAT). The crystal structures were examined using X-ray diffraction (XRD, Rigaku D/Max–2500) with Cu K$_\alpha$ radiation. The electrical and optical properties were ascer-

tained using a Hall-effect measurement method (Ecopia, HMS–3000) and ultraviolet-visible spectroscopy (UV–vis, Perkin–Elmer, Lambda–35), respectively.

### 3.2.2. Electrochemical Measurements

To identify the electrochemical characteristics and EC performance outcomes, a potentiostat/galvanostat system (PGSTAT302N, FRA32M, Metrohm Autolab B.V., The Netherlands) was used. These measurements were conducted via a three-electrode cell with Ag wire (reference electrode), Pt wire (counter electrode), and 1M $LiClO_4$ dissolved in anhydrous propylene carbonate ($C_4H_6O_3$, Aldrich) (electrolyte). Cyclic voltammetry assessments were conducted from $-0.7$ to 1.0 V at a scan rate of 20 mV/s. Moreover, the in-situ transmittance variation during the coloration and beaching process was observed using UV–vis specifically to demonstrate the EC performance capabilities.

### 4. Conclusions

In summary, the surface morphologies of net-patterned FTO films were optimized with the various types of meshes (60 mesh, 40 mesh, and 24 mesh) using the one-pot HUSPD process. We confirmed that the FTO patterning procedure with the proposed method can provide a performance enhancement of ECDs. Here, the 40M-FTO sample exhibited excellent EC performance outcomes, such as a large $\Delta T$ value of 58.7%, fast switching speeds (4.1 s for the coloration speed and 5.9 s for the bleaching speed), and a high CE value of 52.4 $cm^2$/C. These results are mainly due to the (1) superior electrical properties of the dominant (200) plane crystal orientation, and (2) the high transmittance with the optimized net-patterned FTO morphology. These results are highly prospected to lead the current research trend of TCO materials in photocatalysis and ECD applications with excellent optical and electrical properties. In conclusion, optimized net-patterned FTO films using the one-pot HUSPD system can serve as an excellent reference for researchers who focus on TCO materials for photocatalysis and ECD applications.

**Author Contributions:** Conceptualization, S.-J.J., K.-H.K. and H.-J.A.; methodology, S.-J.J., K.-H.K.; validation, S.-J.J., K.-H.K. and H.-J.A.; formal analysis, S.-J.J., K.-H.K. and H.-J.A.; investigation, S.-J.J., K.-H.K.; resources, H.-J.A.; data curation, S.-J.J.; writing—original draft preparation, K.-H.K. and H.-J.A.; writing—review and editing, S.-J.J., K.-H.K. and H.-J.A.; visualization, S.-J.J., K.-H.K.; supervision, H.-J.A.; project administration, H.-J.A.; funding acquisition, H.-J.A. All authors have read and agreed to the published version of the manuscript.

**Funding:** This study was supported by the Research Program funded by SeoulTech (Seoul National University of Science and Technology).

**Conflicts of Interest:** The authors declare no conflict of interest.

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
