# Peer review of "Net-Patterned Fluorine-Doped Tin Oxide to Accelerate the Electrochromic and Photocatalytic Interface Reactions"

_catalysts, doi:10.3390/catal11020249_

Round 1

Reviewer 1 Report

This paper reports patterned FTO films fabricated by spray pyrolysis process and their electrochromic properties. However, the scientific field of this work is coating process rather than catalysis field. The reviewer recommends the authors to resubmit their paper to the journal in the field of thin films or coating technology, since the present form does not satisfactorily provide catalysis insight.

  1. Though the authors suggested that their film is useful for photocatalysis, the reason is unclear, since they did not show any photocatalytic properties. In the reviewer’s opinion, good EC property does not directly imply the good photocatalyst.

  1. Did their film contain stainless mesh? or did they use a mesh just like a mask. Figure 1 is unclear to understand its film structure.

  1. Did their UV-Vis absorption spectra involve the optical absorption or reflection of stainless mesh?

  1. Why was the horizontally supplying system better than vertically supplying system? If the authors claim this, they should show the structure of films, which were prepared by vertically supplying system as a control group.

  1. Electroconductivity would depend on the film thickness. The authors must compare their electroconductive properties under the same film thickness condition.

  1. Why were the carrier concentrations different? They prepared their films by using the same precursor solution (by the same doping density of fluorine).

  1. What was the origin of charge carriers? Electrons supplied by fluorine dopant or those by oxygen vacancies?

  1. Why did their 40M-FTO exhibit the crystal orientation?

  1. Were the reaction temperatures and coating times set at the same conditions for all films?

  1. Abbreviation terms should be defined when they are firstly described in the manuscript. (c.f. 40M-FTO in line 11, OM in line 102, CLSM in line 115 etc.)

  1. The expression of “bare FTO” should be changed as “FTO prepared without mesh” for better understanding.

Author Response

We would like to thank the referee for the constructive comments. We addressed all the points one-by-one and modified the text accordingly when needed. All the changes were highlighted.

Reviewer 2 Report

This paper reports on the optimization of the properties of net-patterned fluorine-doped tin oxide (FTO) films using the horizontal ultrasonic spray pyrolysis deposition method. The novelty of the paper lies in the combination of a simple deposition method with a simple patterning technique. The paper is of interest to the community dealing with transparent conducting oxides; however, at the moment the quality of the paper has to be significantly improved to be accepted in Catalysts. I have several questions and recommendations.

  1. The abstract needs to be improved. Expressions, such as “the highest electrical and optical properties” or “high electrical properties” are not acceptable in scientific papers.
  2. The deposition method should be described in more detail. Figure captions, including the caption of figure 1, are too brief. In an ideal case, the captions should be self-explanatory.
  3. There is just one SEM image of 40M sample. For comparison, it would be useful to have SEM images of other patterned samples as well.
  4. The paper focuses on the patterning of the surface; however, the information on how the mesh was deposited is missing. Moreover, there is no explanation of the phenomena that take place when the surface is patterned. Why is the number of 24 meshes insufficient? What is meant by the statement that the lowest crystallinity was achieved on the bare FTO film due to the low deposition temperature and short deposition time? (LINE 137) From the experimental part, it seems that the temperatures and the deposition times are identical for all the experiments. Is low crystallinity derived from XRD patterns? Is it not just related to the significantly lower thickness of the bare FTO film?
  5. The peaks in diffraction patterns in figure 4 have extremely small amplitudes, which make their interpretation difficult. Again, the caption should explain what key conclusions can be made from the XRD patterns without having to read the text.
  6. How was the Hall measurement performed? In what configuration and by which method were the contacts deposited?
  7. What is the size of the crystallites in the samples? Did you perform a TEM analysis? Why is the thickness of the sample 24M lower than that of 40M and why is it less textured?
  8. Could you explain what would be the ideal crystalline structure of the film to achieve optimum transmittance and conductivity?
  9. Overall, I lack a description of the phenomena that take place during the formation of the films. Without understanding the phenomena the paper brings just an incremental contribution to the scientific community.

Author Response

(The authors gave the same response as above.)

Reviewer 3 Report

In the manuscript, the authors studied the optimization of surface morphology of the net-patterned FTO films using one-pot horizontal ultrasonic spray pyrolysis deposition process. It was found that because of the improved crystallinity and net-patterned surface morphology, the 40 mesh net-patterned FTO exhibited the highest electrical and optical properties. Indeed this is a novel strategy to prepare such transparent conducting electrode with enhanced electrochromic and photocatalytic performance. 

After going through the whole manuscript, I would like to recommend this manuscript to be accepted for publication with minor revision. The authors should address the following comments.

  1. The authors are suggested to combine some of the single figures (e.g. fig 3 & fig 4) to enhance the readability.
  2. The authors should added more insightful discussions in addition to the description of experiment and results.
  3. The authors are suggested to cite more references to attract more readership, e.g.  i) Physical Chemistry Chemical Physics 17 (38), 25333-25341; ii) Physical Chemistry Chemical Physics 12 (38), 11923-11929; iii) Solar Energy Materials and Solar Cells, 95(2), 618-623; iv) Materials, 11(9), 1627.

Author Response

(The authors gave the same response as above.)

Round 2

Reviewer 1 Report

The paper has properly been revised on the basis of the reviewers' comments. The reviewer thinks that the revised version meets the minimum requirements as a scientific journal in the field of catalysis. The paper can be publishable as is.

Reviewer 2 Report

The quality of the paper was significantly increased. Yet, for the future, the authors are recommended to focus on the discussion of physical phenomena during the deposition process to further increase the impact of their papers.